# Health in Preconception, Pregnancy and Postpartum Global Alliance: International Network *Preconception* Research Priorities for the Prevention of Maternal Obesity and Related Pregnancy and Long-Term Complications

**DOI:** 10.3390/jcm8122119

**Published:** 2019-12-02

**Authors:** Briony Hill, Helen Skouteris, Helena J Teede, Cate Bailey, Jo-Anna B Baxter, Heidi J Bergmeier, Ana Luiza Vilela Borges, Cheryce L Harrison, Brian Jack, Laura Jorgensen, Siew Lim, Cynthia Montanaro, Leanne Redman, Eric Steegers, Judith Stephenson, Hildrun Sundseth, Shakila Thangaratinam, Ruth Walker, Jacqueline A Boyle

**Affiliations:** 1Monash Centre for Health Research and Implementation, School of Public Health and Preventive Medicine, Monash University, Level 1, 43-51 Kanooka Grove, Clayton, Victoria 3168, Australia; briony.hill@monash.edu (B.H.); helen.skouteris@monash.edu (H.S.); Helena.teede@monash.edu (H.J.T.); cate.bailey@monash.edu (C.B.); Heidi.bergmeier@monash.edu (H.J.B.); cheryce.harrison@monash.edu (C.L.H.); siew.lim1@monash.edu (S.L.); ruth.walker@monash.edu (R.W.); 2Warwick Business School, Warwick University, Coventry CV4 7AL, UK; 3Monash Partners Advanced Health Research Translation Centre, Locked Bag 29, Clayton, Victoria 3168, Australia; 4Monash Health, Melbourne, 246 Clayton Road, Clayton, Victoria 3168, Australia; 5Centre for Global Child Health, The Hospital for Sick Children, Peter Gilgan Centre for Research and Learning, 686 Bay Street, Toronto, ON M5G 0A4, Canada; jo-anna.baxter@sickkids.ca; 6Department of Nutritional Sciences, Medical Sciences Building, University of Toronto, 1 King’s College Circle, Toronto, ON M5S 1A8, Canada; 7Public Health Nursing Department, University of Sao Paulo, 419 Cerqueira Cesar, Sao Paulo 05403000, Brazil; alvilela@usp.br; 8Department of Family Medicine, Boston University School of Medicine, 771 Albany St, Boston, MA 02118, USA; bjack@bu.edu; 9Barts Research Centre for Women’s Health (BARC), Women’s Health Research Unit, Centre for Primary Care and Public Health, Blizard Institute, Barts and The London School of Medicine and Dentistry, 58 Turner Street, London E1 2AB, UK; laurajjorgensen@gmail.com (L.J.); s.thangaratinam@qmul.ac.uk (S.T.); 10Wellington-Dufferin-Guelph Public Health, 160 Chancellors Way, Guelph, ON N1G 0E1, Canada; cynthia.montanaro@wdgpublichealth.ca; 11Reproductive Endocrinology and Women’s Health Laboratory, Pennington Biomedical Research Center, 6400 Perkins Rd, Baton Rouge, LA 70808, USA; leanne.redman@pbrc.edu; 12Department of Obstetrics and Gynaecology, Erasmus Medical Centre—Sophia Children’s Hospital, Wytemaweg 80, 3015 CN Rotterdam, The Netherlands; e.a.p.steegers@erasmusmc.nl; 13Institute of Women’s Health, University College London, EGA Institute for Women’s Health, 74 Huntley St, London WC1E 6AU, UK; judith.stephenson@ucl.ac.uk; 14European Institute of Women’s Health, 33 Pearse Street, Dublin 2, Ireland; h.sundseth@gmx.de

**Keywords:** preconception care, obesity prevention, lifestyle behaviours, consensus, research priorities

## Abstract

The preconception period is a key public health and clinical opportunity for obesity prevention. This paper describes the development of international preconception priorities to guide research and translation activities for maternal obesity prevention and improve clinical pregnancy outcomes. Stakeholders of international standing in preconception and pregnancy health formed the multidisciplinary Health in Preconception, Pregnancy, and Postpartum (HiPPP) Global Alliance. The Alliance undertook a priority setting process including three rounds of priority ranking and facilitated group discussion using Modified Delphi and Nominal Group Techniques to determine key research areas. Initial priority areas were based on a systematic review of international and national clinical practice guidelines, World Health Organization recommendations on preconception and pregnancy care, and consumer and expert input from HiPPP members. Five preconception research priorities and four overarching principles were identified. The priorities were: healthy diet and nutrition; weight management; physical activity; planned pregnancy; and physical, mental and psychosocial health. The principles were: operating in the context of broader preconception/antenatal priorities; social determinants; family health; and cultural considerations. These priorities provide a road map to progress research and translation activities in preconception health with future efforts required to advance evidence-translation and implementation to impact clinical outcomes.

## 1. Introduction

Obesity is a leading public health problem, with childbearing a key driver of obesity development in women [1]. High preconception body mass index (BMI), alongside excess gestational weight gain and postpartum weight retention, are significant and independent contributors to rising maternal obesity and associated health risks [2,3]. Long-term lifestyle and medical treatment of obesity is largely ineffective and unable to curb associated adverse health outcomes; prevention is essential.

A recent Lancet series highlighted the importance of health and wellbeing on preconception health, with actions required varying across the lifecourse and most targeted when actively planning a pregnancy [4]. The preconception period can be conceptualised in three ways to ensure that all preconception populations are captured for intervention: (1) in the days to weeks before embryo development—a biological perspective; (2) in the weeks to months before pregnancy for individuals with a conscious intention to conceive; and (3) as a public health perspective with the longer period of months to years required to address preconception risk factors [4]. In particular, nutrition and lifestyle were areas requiring attention globally, due to the high prevalence of poor nutrition and obesity in women of reproductive age. Preconception is now recognised as a period of opportunity for lifestyle intervention because of the potential impact on future generations [4,5,6]. Preconception obesity is linked to adverse health outcomes in offspring such as obesity, coronary heart disease, stroke, type 2 diabetes, asthma, poorer cognition, and neurodevelopmental disorders [7]. Planning pregnancy allows for more time to take actions to improve health, protect fertility, and increase the chance for healthy maternal and birth outcomes [4]. However, pregnancy planning varies significantly; 30 to 50% of pregnancies are unintended in high-income countries and even more in middle- and low-income countries [8].

A 2006 report of the US Centers for Disease Control/Agency for Toxic Substances and Disease Registry that outlined recommendations to improve preconception health and health care was one of the first publications to recognise obesity as a risk factor for preconception health [9]. More recently, international and national bodies such as the World Health Organization (WHO) [10], the UK National Institute for Health and Care Excellence [11], Health Canada [12], and the US Institute of Medicine [13] have identified the preconception period (with postpartum considered a key inter-conception phase) as a key opportunity for obesity prevention. Despite this, very few international or national guidelines on weight management or weight-related lifestyle behaviours have specific recommendations for women in the preconception period [11]. A significant contributing factor is the inadequate knowledge base at the empirical, applied, translational, and implementation levels to generate quality guidelines. Such research is essential to inform development of policy directives and guidelines for practice at both community health and primary care levels to deliver public health impact. In particular, obesity prevention efforts targeting the preconception period specifically, with both individual and systems level intervention, are scarce [14]. Existing networks focusing on preconception health and care that conduct important work in this field (e.g., the “PrePreg Network”) are not focused specifically on obesity prevention. There is also a lack of coordination in efforts and strategic research to address key evidence gaps.

Building on previous success in developing international and national alliances in Health in Preconception, Pregnancy and Postpartum (HiPPP) [15] and polycystic ovary syndrome (PCOS) [16], a leading endocrinologist, gynaecologist, and psychologist at the Monash Centre for Health Research and Implementation (MCHRI), Monash University, convened an international forum in Prato, Italy in September 2018. The forum was part of a strategy to improve stakeholder engagement in the field. It included a network panel with invited experts and consumers from across the globe to collectively work towards optimising preconception and pregnancy healthy lifestyle and prevent maternal obesity and related short- and long-term complications. Forum attendees formed the HiPPP Global Alliance. This paper aims to describe the activities of one of the key forum objectives: to develop agreed upon international preconception priorities to address the global issue of poor lifestyles in reproductive aged women, specifically for the prevention of maternal obesity and related pregnancy clinical outcomes. To our (i.e., the HiPPP Global Alliance) knowledge, no such existing priorities have been established to guide research and translation activities. The process of establishing pregnancy priorities for the prevention of maternal obesity is presented elsewhere in this issue [17]. 

## 2. Method

### 2.1. Process

A modified Delphi process and Nominal Group Technique were used to determine the preconception priorities [18,19]. This process has been used previously to establish priorities for data driven healthcare improvement and cancer care and was specifically adapted to address research goals in this context [19]. The multistep process is outlined in Figure 1 and occurred in three phases: inputs, ranking, and output, with the process spanning before, during, and after a workshop held at the HiPPP international forum. The workshop was four hours in duration and was facilitated by HT with the assistance of two early career researchers (BH and CB).

### 2.2. Phase I. Inputs

#### 2.2.1. Identification of Forum Participants: Experts and Consumer Representatives

Thirteen stakeholders of international standing in their respective fields with regards to preconception and pregnancy health were invited, representing geographic diversity, and variations across clinical or academic discipline. Two consumer representatives (also known as consumer and community involvement (CCI) or patient and public involvement (PPI) members; LJ and HiS) were also invited from established non-governmental women’s health and consumer representative organisations and had received training through these organisations regarding participation as consumer experts in research activities. Six early career researchers with relevant expertise were also invited (BH, CB, HJB, CLH, SL, and RW); their input was weighted so their combined votes were equivalent to one international expert. In total, 21 participants were invited to the forum. Two of the invitees were unable to attend but nominated a replacement who was able to attend. One invitee had planned to attend but was unable to at the last minute; hence 20 participants attended (95%).

#### 2.2.2. Research Priority Areas

The research priority areas were developed based on a comprehensive systematic review of international and national clinical practice guidelines (CPGs) on preconception and pregnancy care and the WHO recommendations on preconception and pregnancy care [20,21]. The aim of the CPG review was to evaluate CPGs for weight management and associated healthy lifestyle behaviours across preconception, pregnancy and postpartum to consolidate and streamline content to optimise weight management strategies in this setting. 

#### 2.2.3. Priority Setting Framework

A framework for application during the priority setting process was adapted from the Australian Policy Prioritisation Framework, which was successfully used for the development of Australian national priorities in data driven healthcare improvement and women’s health [22,23]. The framework offered nine criteria (9Ps) for priority assessment: (i) prevalence or burden, (ii) prevention, (iii) position, (iv) provision, (v) potential, (vi) participation, (vii) policy, (viii) proposed strategy, and (ix) proposed transformation. The 9Ps were reviewed and endorsed by HiPPP members and were used to draw participants’ attention to research and evidence-translation gaps when establishing the priorities (Table 1).

### 2.3. Phase II. Ranking

#### 2.3.1. Round 1: Pre-Workshop Ranking

One month prior to the workshop, the 20 forum participants (plus the one expert who did not attend the forum) were sent via email 12 preconception priorities for consideration. Using a modified Delphi format, each participant was asked to rank the priorities, with 1 being the highest ranked priority and the remaining in ascending order, and return their ranking via email. Participants were able to suggest additional priorities that were not listed. Mean ranking scores were computed for each priority, with lower scores representing higher priority.

#### 2.3.2. Rounds 2 and 3: Workshop Group Discussion and Independent Voting

At the workshop, Nominal Group Technique was employed for consensus development [18], led by an endocrinologist (HT) experienced in international network development and priority setting [15]. Firstly, in groups of four to five, participants discussed the Round 1 rankings. Then, groups considered whether any priorities could be consolidated, amalgamated, added, or removed. This process was followed by a facilitated whole group discussion to consolidate and integrate group inputs, with care taken to ensure all participants’ voices were considered and captured.

Once a consolidated list of priorities was agreed upon, participants were asked to independently and confidentially rank the new list of priorities for preconception. Ranking scores were summed to create a total score for each priority. Participants were provided with the Round 2 scores/ranking and were directed back to their groups to discuss the priorities with reference to the 9Ps priority setting framework. Considering the 9Ps, participants were asked to rank the priorities to derive a final priority list.

### 2.4. Phase III. Output

#### 2.4.1. Consensus Development of Priorities

Participants’ individual final vote scores were totaled, ranked and presented to the group. Facilitated discussion regarding minor modifications occurred, with a majority vote used to approve any proposed changes. Participants were asked to form a consensus on the number of top priorities as the focus of future research opportunities. The final workshop activity involved small group discussions on specific research, practice, and policy gaps and initial plans of action for the priorities. Subsequently, the ideas were shared with the whole group and shaped further through facilitated group discussion.

#### 2.4.2. Post Workshop Consultation and Collaboration

After the workshop, the priorities and resultant research, practice, and policy gaps were circulated electronically among the expert and consumer stakeholders for final consultation.

## 3. Results

### 3.1. Phase I. Inputs

#### 3.1.1. Participants

Participants included experts from multiple disciplines, including medicine, nursing, psychology, health economics, allied health, epidemiology, applied health services research, paediatrics, global health, as well as consumer advocates. Participants were from across five continents, including the countries of Australia, Belgium, Brazil, Canada, The Netherlands, South Africa, UK, and US.

#### 3.1.2. Research Priority Areas

From the CPG review, 20 CPGs that included recommendations for weight management, diet and physical activity across preconception (*n* = 2), pregnancy (*n* = 8), postpartum (*n* = 2) or a combination (*n* = 8) were used to inform priority setting, along with areas identified in the WHO preconception care policy brief [20]. The initial 12 priorities for preconception lifestyle intervention (in no particular order) were dietary interventions, physical activity, weight management, diabetes and chronic disease, mental health, nutritional supplementation, sexually transmitted infections (STIs) and blood-borne viruses (BBVs), vaccine-preventable diseases, genetic conditions, infertility/sub-fertility, planned pregnancy, and tobacco and substance use.

### 3.2. Phase II. Ranking

#### 3.2.1. Round 1: Pre-Workshop Ranking

Pre-workshop rankings are presented in Table 2. Several additional priorities were also suggested: reproductive and obstetric history, other infectious diseases, healthy relationships and violence against women, and the health of the partner or spouse.

#### 3.2.2. Round 2: Workshop Sense-Making, Group Discussion and Independent Voting

Throughout discussion, preconception was defined according to all three definitions proposed in the 2018 Lancet series, recognising that different approaches are required depending on the definition used [4]. Regarding the individual priorities, it was agreed that ‘dietary interventions’ and ‘nutritional supplementation’ could be combined into a ‘healthy diet and nutrition’ priority that included folic acid supplementation and food security. The ‘diabetes and chronic disease’ priority was modified to become ‘pre-existing medical conditions’ that also included chronic disease. ‘Planned pregnancy’ and ‘infertility/sub-fertility’ were combined into ‘planned pregnancy including awareness and optimising fertility’. ‘Sexually transmitted infections and BBVs’ was modified to ‘infections’, which included vaccine-preventable diseases, STIs, BBVs, and other infectious diseases. An additional priority, ‘healthy relationships’ was added. The final rankings after Round 2 voting are presented in Table 2.

It was also agreed that an overarching set of principles was required to be considered against all priorities. These related to factors that were common across all priorities and became repeated themes in the discussion. It was continually raised in discussion that issues related to racial, ethnic, or income bias must not be ignored. The principles were: (a) context of broader preconception/antenatal care priorities; (b) social determinants of health; (c) health of families; and (d) cultural considerations (See Box 1).

Box 1Final consensus for preconception priorities and overarching principles.
**Preconception Priorities**
Healthy diet and nutrition, including
Folic acid supplementationFood securityWeight managementPhysical activityPlanned pregnancy including awareness and optimising fertilityPhysical, mental and psychosocial health, including
Chronic disease including diabetes, hypertensionPre-existing pregnancy conditions

**Overarching Principles**
Context of broader preconception/antenatal care prioritiesSocial determinants of healthHealth of familiesCultural considerations


#### 3.2.3. Round 3: Workshop Group Discussion and Independent Voting

After considering the new lists of priorities against the 9Ps, the final independent ranking vote generated the top eight preconception priorities. Mental health was noted to be low on the list and discussion on the role mental health can play in the attainment of healthy lifestyle behaviours ensued. A majority vote agreed that mental health should be moved from rank seven to six. It was then discussed that mental health would fit better when combined with physical health and so was combined with ‘pre-existing medical conditions’ and became ‘physical, mental and psychosocial health’. Round 3 rankings are presented in Table 2.

### 3.3. Phase III. Ouput

#### 3.3.1. Consensus Development of Priorities

A consensus was achieved by the experts that the top five preconception priorities would ideally form the basis of future research and evidence-translation activities to prevent maternal obesity and related pregnancy and long-term complications (see Box 1). Within some priorities, key points were noted where special attention was required (e.g., folic acid within nutrition). 

The final group discussion identified specific research, practice, and policy gaps for preconception that would help address the identified priorities. It was identified that significant formative research and evidence synthesis are required, including further understanding of preconception populations (e.g., defining the target groups, understanding their physical, behavioural and psychosocial characteristics), the impact of preconception weight loss on pregnancy outcomes, and co-designed interventions and clinical trials. Pragmatic, multinational implementation trials are needed for preconception lifestyle interventions; collaboration is key to achieving these goals. A biopsychosocial context should be integrated, with policy directives aligned. Personalised medicine, population health, and systems level thinking should be integrated to address the priorities. These goals must be included within multifaceted interventions that incorporate tool kits, training (e.g., for health professionals), and real-world implementation strategies, with a focus on eHealth. Once the evidence-base is sufficient, policy directives are needed to ensure adequate translation across the healthcare and public health systems, including government, guidelines, and health professional training and development.

#### 3.3.2. Post-Workshop Consultation and Collaboration

After a post-workshop consultation opportunity, all participants agreed with the research, practice, and policy gaps identified. The HiPPP Global Alliance members committed themselves to working collectively and collaboratively to address the preconception priorities and achieve the goals outlined for the prevention of maternal obesity and related complications.

## 4. Discussion

International preconception research and translation priorities for healthy lifestyle and the prevention of maternal obesity and related pregnancy and long-term complications were identified by the multidisciplinary HiPPP Global Alliance. A multistep, transparent, modified Delphi and Nominal Group Technique consensus development process was applied. Five preconception priorities and four overarching principles were identified. 

The highest ranked preconception priority for maternal obesity prevention was ‘healthy diet and nutrition’. Given that diet and nutrition are key drivers of weight-related outcomes, these need to be addressed pre-pregnancy, in both long-term (lifecourse) and short-term (shortly before pregnancy) approaches [4]. Folic acid supplementation and food security were mentioned specifically within this priority, notwithstanding other important factors associated with diet and nutrition preconception, including advancing our understanding of the dietary intakes of preconception populations, and exploring how preconception improvements in diet can impact on pregnancy outcomes and beyond.

Weight management was the second highest ranked priority. Weight management was defined as interventions targeting individuals at an unhealthy weight and may include strategies beyond diet and physical activity, for example bariatric surgery. Research is required to explore these approaches and their potential impact on future pregnancy outcomes. Physical activity was the third ranked priority with future research required to enhance our knowledge of both the physical activity of preconception women and how this can be optimised before pregnancy. This is pertinent given the declining levels of physical activity in women of reproductive age and the further decline seen in pregnancy [24].

The fourth priority was planned pregnancy, including awareness and optimisation of fertility. Work is needed to improve not only pregnancy planning, but to improve awareness of health behaviours in all women of reproductive age regardless of pregnancy intention [4]. This will require both targeted and broader public health initiatives [4]. The benefits of weight loss in women who are above healthy weight, or have other specific conditions, such as PCOS, have been well documented in improving cycle regularity and fertility [25]. However, further exploration is needed on the impacts of significant weight loss and of different types of interventions just before pregnancy on fertility, maternal, fetal, and infant outcomes.

The fifth and final ranked priority for targeted research was physical, mental, and psychosocial health, recognising the interrelationships between those domains of health. This priority includes prevention and management of chronic diseases associated with obesity and lifestyle, such as diabetes and cardiovascular disease, that also impact pregnancy outcomes and beyond. Pre-existing pregnancy conditions (including medical history from previous pregnancies) are also included here, requiring management in the preconception phase to ensure they are adequately controlled into subsequent pregnancies. Research is needed to improve understanding of how to address mental health concerns and medication management before pregnancy so as to positively impact lifestyle and weight outcomes during this life phase. Overall physical, mental, and psychosocial health are all seen as crucially important within this priority.

Four overarching principles were identified as being applicable to each research priority area. All research in this field should take into consideration context of all the preconception, as well as any pregnancy [17], research priorities—there is much interplay between these priorities. Women, health professionals, and researchers should co-design and partner in future work and ensure that evidence, knowledge, and experiences across the broader context meet stakeholder needs. The health of families is also an essential consideration given that preconception health can have direct impacts on offspring health [26] and because family provides context within which preconception women live [27]. Social determinants of health must also be considered given the inequalities in health indicators and in the access to health services across the socio-economic spectrum [28]. This includes addressing issues such as poverty, access to education, family violence, and other factors, which may limit women’s agency for health-promoting behaviours and pregnancy planning [29]. Cultural considerations across culturally and linguistically diverse populations and other social and economic contexts globally are imperative to ensure equity (for example in research conduct, intervention delivery and policy generation) and to avoid further increasing disparities in outcomes. Examples include ensuring the right questions are asked, diverse populations are included in research, research is conducted in an appropriate manner, research is funded to ensure women who speak different languages are included, and translation, implementation and policy take into account cultural diversity and needs. Applying an ecological systems approach when addressing the identified priorities will help ensure that domains across the spectrum from the individual woman herself, through family and home; work, school and peers; community; industry and government; to culture and society are addressed [30]. Examples of activities that may address these priorities in the preconception period include involving partners in research or implementing policies that increase access to healthcare services, such as preconception health screening, and accessible and affordable health education for marginalised communities.

As part of workshop discussion, key research, practice, and policy gaps were identified to further develop the evidence-base in preconception health, including evidence synthesis, co-designed interventions, real world trials, and implementation research. For example, studies (such as longitudinal designs and birth cohorts) are needed to improve our basic understanding of mechanisms, pathways, biological drivers, and impacts on outcomes, including outcomes for the next generation. Evidence synthesis is required for all levels of evidence including observational studies, randomised controlled trials, and pragmatic implementation trials, when a sufficient number of trials have been evaluated. Co-design techniques (i.e., collaborative processes where the key stakeholders such as patients, consumers and health professionals work together to develop interventions that meet the expectations and needs of the target audience [31]) are paramount to the design of effective interventions that are based on stakeholder partnership, and that can be implemented in a cost-effective manner at scale. As part of this, consumer engagement is essential early in the research process and across the research translation pathway. Consumer engagement refers to the active partnership between the researchers and those affected by the research (e.g., recipients of a health service) in the research process, as opposed to having research conducted to or for them [32]. Consumer involvement may take many forms during all stages of the research cycle, from that of research participant to research partner [33]. Finally, translation pathways must be identified, which will include incorporating system level approaches, health professional training, and adaptation of policies based on the new evidence as it is generated. In order to achieve these aims, forum members agreed the HiPPP Global Alliance should work towards four key short- and medium-term goals identified as forum objectives: (i) generate a high-level position statement to capture and integrate international and national guidelines on lifestyle modification across the preconception, pregnancy and postpartum periods; (ii) develop an agreed consumer engagement and advocacy strategy; (iii) develop agreed workforce capacity building strategies; and (iv) develop capacity in early career researchers in the HiPPP fields. The HiPPP Global Alliance are also committed to actively seeking collaborative opportunities both within and external to the alliance to capitalise on knowledge, expertise, resources and funding opportunities, and to publish, share, and implement knowledge gains across academic, policy, and health sectors [34]. 

The priorities were generated through a robust process that took into consideration existing international and national guidelines in preconception and pregnancy care. Consumer representatives were actively involved in the process. The consensus development technique employed was empirically supported and minimised bias by potential dominant participation of some individuals or groups [19]. Participants were diverse in discipline, geographic location, and foci of research activities and represented primary care, community health, and public health, conducting research across preconception, pregnancy and the postpartum periods. Many of the preconception research priorities can be promptly addressed, and in some cases, work is currently underway by HiPPP Global Alliance members (see Box 2). Limitations include the fact that not all invited participants took part in the process, including participants from Asia, many experts who could have contributed knowledge in the field may not have been invited in order to balance global input, not all disciplines were equally represented at the workshop, and that the Policy Prioritisation Framework did not explicitly identify issues related to racial, ethnic or income bias. Finally, the social determinants of equity were not explicitly raised in the overarching principles, however they cannot be ignored. Global issues such as racism and systemic oppression [35,36] must be addressed in all health research, including the preconception priorities identified here.

Box 2HiPPP Global Alliance research and evidence-translation activities to address preconception priorities for maternal obesity prevention.
**Completed activities that contribute to the priorities**
Establishment of Katie’s Team, a women’s health research patient and public advisory group for East London.Publication of a 2018 Lancet series that described the different contexts in which we can address nutrition and lifestyle preconception [4].Validation of the London Measure of Unplanned Pregnancy in the Australian context and cultural adaptation and validation in Brazilian Portuguese (http://www.lmup.com/). Development and evaluation of eHealth platforms for preconception health and care:
○The virtual conversational agent, Gabby, which screens for 100 risk factors in preconception care, and has been designed to meet the gap between the number and availability of clinicians and what is needed in preconception care in the US [37].○A Canadian preconception health risk assessment tool intended to be used in the primary care setting to improve preconception care, adapted from Gabby (wdgpublichealth.ca/preconception-health). Both the US and Canadian programs are context specific and are undergoing evaluation.○Development of the Dutch Preparing for Pregnancy website, which offers an internet questionnaire for risk assessment in preconception care [38].○The mHealth ‘Smarter Pregnancy’ platform, a personal online coaching program for couples contemplating pregnancy [39].Publication of our first paper as an Alliance for the HiPPP Global Alliance, highlighting that we now have a clear and imperative call to action to consolidate and advance current evidence into practice and policy for the prevention of maternal obesity [34].

**Ongoing activities**
Further defining the preconception population (beyond the definitions presented in Stephenson et al. [4]) and deepening our understanding of the lifestyle behaviours and mental health of women before pregnancy. Understanding the social determinants and disparities in perinatal health, which, for example, contribute to worse outcomes for pre-term delivery, small-for-gestational age and stillbirth [28].Conducting an intervention trial aiming to better understand the effect of preconception multiple micronutrient supplementation and life-skills-based education compared to the standard of care, among adolescent and young women in rural Pakistan. Outcomes will capture the health of both those who do and do not become pregnant within the context of the trial [40,41].Conducting an intervention to evaluate whether training health professionals on preconception health will improve delivery of preconception services in primary health care facilities in Sao Paulo, Brazil.In pregnancy, collaborative work of individual patient data meta-analysis and comprehensive systematic reviews of lifestyle intervention to reduce gestational weight gain have highlighted that generation of evidence needs to move from randomised controlled trials to pragmatic implementation trials sooner [42]; lessons learned in this space will be applied preconception.Applying our recent work developing our understanding of intervention strategies, behaviour change techniques, and implementation characteristics in lifestyle interventions for postpartum [43] women to preconception applications.Conducting an updated review of preconception and pregnancy guidelines for lifestyle modification in pregnancy.


## 5. Conclusions

An international priority setting process including a workshop was completed by the HiPPP Global Alliance to establish research and evidence-translation priorities to optimise healthy lifestyle and prevent maternal obesity and related short- and long-term complications with a focus on the preconception period. Five key preconception research priorities were identified to address this issue collaboratively, both within and beyond the HiPPP Global Alliance, focusing on healthy diet and nutrition, weight management, physical activity, planned pregnancy and physical, mental, and psychosocial health. In addition, four overarching principles were identified across all research activities. The HiPPP Global Alliance is currently working towards implementation of the priorities identified, however many gaps remain, and significant work is needed across the broader research community to advance preconception research towards translation and implementation. The HiPPP Global Alliance is aware of other research teams that focus on these priorities and welcomes additional opportunities for collaboration.

## Figures and Tables

**Figure 1 jcm-08-02119-f001:**
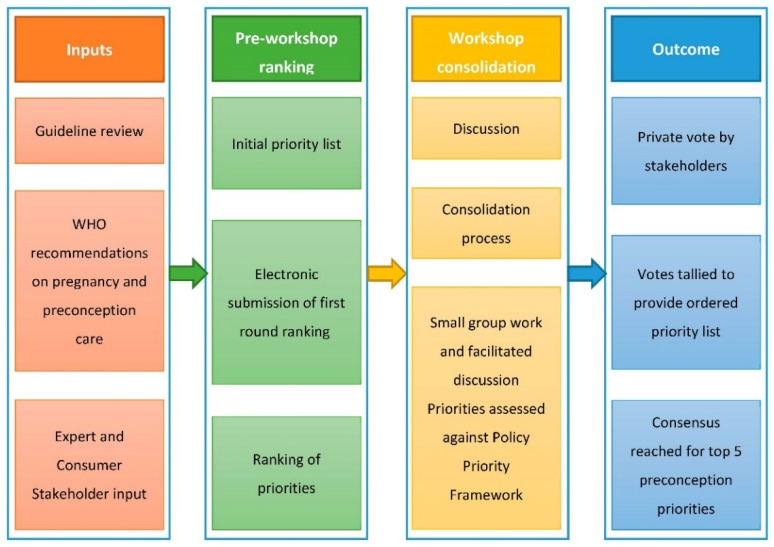
The consensus development process for preconception research and translation priorities.

**Table 1 jcm-08-02119-t001:** Priority setting framework (9Ps).

Criteria	Definition
Criteria 1. Prevalence or burden attributable to the proposed problem	Consider the prevalence or attributable burden of the problem and its implications/complications. Is the problem a significant issue for the community, health system and key stakeholders?
Criteria 2: Prevention	Is there potential to prevent the problem, including complications or secondary impacts, in the general population or in a specific vulnerable target cohort?
Criteria 3: Position	Consider the geographical issues around the problem and the location of services/expertise. Are there inequities that can be improved through this initiative? Is there potential to improve health outcomes for the general population and/or regional populations and/or specific vulnerable target cohorts?
Criteria 4: Provision	Does the current approach or system align with evidence-based best practice? Is the current approach designed to deliver the best possible community health outcomes and health care system? Is there a clear gap to address in the area proposed?
Criteria 5: Potential	Is there a strong rationale/evidence base for the potential for improvement in patient outcomes and health system advancement through this initiative?
Criteria 6: Participation	Is a collaborative approach critical to success? Are there clear drivers for stakeholders to engage and collaborate? Are there existing relationships between stakeholders that can be leveraged to drive improvement and change?
Criteria 7: Policy	Does the problem or the potential solution align with current policy directions at a local, state, national or international level?
Criteria 8: Proposed Strategy	Does the proposal align with the purpose of the Health in Preconception, Pregnancy and Postpartum strategic alliance?
Criteria 9: Proposed Transformation	Will addressing this problem or taking this approach collaboratively support the development of an improved health system and health outcomes?

**Table 2 jcm-08-02119-t002:** Preconception rankings after round 1, 2 and 3.

	Round 1	Round 2	Round 3		
Preconception Priority	Ranking	Ranking	Ranking	Mean (SD)	Median (IQR)
Healthy diet and nutrition Folic acid supplementation Food security	1, 8 ^†^	1	1	1.6 (0.9)	1.0 (1.0)
Physical activity	3	2	2	2.9 (1.3)	3.0 (1.0)
Weight management	2	3	3	3.4 (0.9)	4.0 (1.0)
Planned Pregnancy—awareness and optimising and fertility	5, 10 ^†^	4	4	4.4 (2.3)	4.5 (2.8)
Pre-existing medical conditions * Chronic disease including diabetes, hypertension Pre-existing pregnancy conditions	4, 12 ^†^	5	5	4.5 (2.1)	4.5 (3.2)
Substance use (including alcohol and tobacco)	6	7	6	6.4 (1.4)	6.0 (1.8)
Mental health *	7	6	7	5.6 (1.8)	6.5 (2.0)
Infections Vaccine-preventable diseases STIs and BBVs Other infectious diseases	9, 11 ^†^	9 ^^^	8	8.6 (0.5)	9.0 (1.0)

* After the Round 3 ranking, it was discussed and decided via majority vote that mental health be combined with pre-existing medical conditions to form one priority of ‘physical, mental and psychosocial health’. ^†^ Priorities were ranked individually in round 1, and then combined during the sense making process such that some separate priorities were amalgamated. ^^^ In Round 2, the 8th ranked priority was ‘healthy relationships’ but it was decided to this was covered in the overarching principles and it was removed from the ranking list.

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
