# Peer review of "Health in Preconception, Pregnancy and Postpartum Global Alliance: International Network Preconception Research Priorities for the Prevention of Maternal Obesity and Related Pregnancy and Long-Term Complications"

_jcm, 2019, doi:10.3390/jcm8122119_

Round 1

Reviewer 1 Report

Summary:

Overall, a timely review which will support planning and funding of studies based on preconception care and can help focus research efforts and minimise waste. The methods used are appropriate but require clarifications for some steps. The authors investigated research priorities in preconception health and identified 5 priorities and 4 overarching principles spanning several domains which have been described well. However, providing some clarity/ details on the recommendations/ considerations for future research and public health policies is suggested to increase the impact of the report (detailed comments below). The article is generally well-written.

Introduction:

Would be useful here to have a note on the definition of preconception group considered here - was it the general reproductive age group for all women or the WHO definition of preconception period? This would also be helpful to understand which target group was explored during the review process of this study.

Methods:

Overall the methods used are appropriate for the research question but some clarifications are needed. Adequate information on the priority setting framework has been provided. For the pre-workshop ranking survey, numbers have not been provided for participants. How many participants were invited and what proportion of these participants attended the workshop? Was the invitation sent only to the participants/ members of the HiPPP forum, if yes, could this have created a bias? What was the aim of the systematic review? Please provide a reference if this is a separate paper, or information can be summarised in the Results section on the key findings. Did the consumer representatives include any lay members or PPI participants? It is mentioned that all authors were workshop attendees. Please add the total number of attendees of the workshop. Please provide details if ethics approval was in place for data collection. 

Results:

What were the results of the policy guidelines review? Please provide some information on how the review helped in constructing the 12 priority areas for the Delphi survey. For e.g. were the 12 priority areas a list of all the domains that emerged from existing literature from the review? The authors have described in adequate detail how mental health became part of the priority-5. However, I think it would be helpful to highlight how mental health and wellbeing should not be weakened as a priority because of this, considering the increasing evidence for the importance of preconception and interconception mental health. Line 268 - any examples for the policy gaps? 270 - “Further understanding of preconception populations” - do you mean defining target groups clearly?

Discussion:

Line 321 - 333 for the overarching themes: Issues such as the social determinants of health are certainly crucial, hence the implications and suggestions from the workshop can be fleshed out further in this paragraph. Did the group have any specific/actionable recommendations on how the overarching principles can be embedded in research and public health practice (or on the current state and how it can be improved)? As the research question was formulated to identify the gaps and priority areas, the methods used were appropriate (review, Delphi survey and NGT). However, results of the review have not been sufficiently provided in the report. Also, it is mentioned as a forum objective - 1 Line 347. This is a bit confusing - please clarify if the review of preconception guidelines was conducted before the Delphi survey, or was it for intervention studies only? Line 334-343 some of the points discussed in the workshop are mentioned here and authors acknowledge that co-design and consumer engagement/ stakeholder partnership is important. Please clarify what you mean here - Do you mean mixed methods and qualitative research? What was the discussion (if any) during the Delphi process/ group meeting about birth cohorts and longitudinal data for preconception studies and the lack of it? Research has highlighted the lack of longitudinal data and complex interventions in the preconception period (e.g. in the Lancet series papers referenced in this report). It would be interesting to know if the HiPPP alliance meeting considered the life-course model and long-term health as a key element in research design which some recent studies are incorporating (Bull F, Willumsen J. Evidence to prevent childhood obesity: The continuum of preconception, pregnancy, and postnatal interventions. Obesity Reviews. 2019) The abstract and introduction mention developing the roadmap to evidence translation, but this has not been adequately discussed in this section. The recommendations suggest academic and some public health measures. What were the policy measures identified? BOX 2 - is a bit confusing - can be rearranged and summarised further. For e.g. some of the references for the activities of the HIPPP global alliance are dated before the formation of the alliance in 2018. Perhaps be they can be separated into completed and ongoing activities? Please consider revising or arranging in chronological order with brief bullet points in a uniform fashion.

Reviewer 2 Report

Thank you for the opportunity to review this work. As someone who has spent well over a decade leading work in this arena I am pleased to see people continue to work in this field. I would have liked to have seen in the introduction a bit more information about the work that is underway in other settings - for example, there is a robust body of work in the US and there are networks and a recent European gathering that weren't mentioned. It would be helpful to be inclusive of this work. I think the methods in the study we're fine and well described. I found the Australian Policy Prioritisation Framework to be a good tool for the paper. The Delphi process has been used in many other settings and seems to have been applied appropriately here. The element that is missing from the review are questions around equity - particularly for women who are racial / ethnic minorities in their perspective countries. The Australian framework doesn't identify issues related to racial/ethnic/income bias. This is a pressing issue globally for many reasons, most important that preconception health if not focused on the women who need it most can further increase disparities in outcomes. I would like to see a table that would provide a better description of the participants - what is the demographic representation beyond just nationality? How many consumers were there? How were they prepared to interact in this environment? How many people overall were there? When the pronoun "our" is used who does that include? I would agree that the social determinants of health are important but even more important are the social determinants of equity. I think that considerations around racism and systemic oppression are as important as the social determinants of health and should be named explicitly. Box 2 does not mention the work underway in the US. Some of the bullet points need more description to make sense. I think it is great that the Alliance exists but as a reader and an advocate in this space this paper feels exclusive. The manuscript is well written and reflects a very important area of work that needs greater visibility, research and funding.
